# ReTaMTSF: Retrieval-Based Multimodal Framework for Multivariate Time Series Forecasting

## Abstract

The rapid advancement of Large Language Models (LLMs) has ushered multi-variate time series forecasting (MTSF) into a transformative era through the integration of natural language. Despite effectiveness of recent language-integrated TSF approaches, they originally stem from engineering intuition, lacking theoretical grounding, and entail considerable manual effort. Moreover, given the importance of inter-channel correlations in MTSF task, current MTSF methods either superficially investigate the intrinsic relations among time series channels or rely heavily on expert knowledge to predefine them, both are with limited flexibility. To address these challenges, we provide an information-theoretic analysis of the role of textual information in augmenting TSF and propose Re-TaMTSF, an MTSF paradigm that automatically aligns and incorporates exogenous text with time series while adaptively capturing inter-channel correlations. We further introduce ReTaMForecaster, a baseline model for ReTaMTSF, and validate its effectiveness through extensive experiments on multimodal MTSF benchmarks spanning diverse domains. ReTaMForecaster achieves state-of-the-art or second-best performance in more than half of the benchmarks and forecasting horizons, with mean squared error (MSE) reductions of up to 74% compared to the best baseline, thereby demonstrating the soundness of the proposed framework with substantial manual effort reduction. Code is available at
`https://anonymous.4open.science/r/ReTaMTSF-CC0A/`.

## 1 Introduction

Time series (TS) are a ubiquitous type of data in both daily life and various engineering practices. In particular, multivariate time series (MTS) data contain underlying information that characterizes system dynamics and reveals the interrelationships and operating mechanisms of complex systems. Owing to their significant research value, MTS forecasting problem has attracted extensive attention from researchers across a wide range of domains, including earth sciences (Karpatne et al., 2019), transportation (Jin et al., 2024a), energy (Zhu et al., 2024), healthcare (Harutyunyan et al., 2019), environmental studies (Xia et al., 2023), finance (Xu et al., 2025), and so on.

To achieve accurate prediction of MTS, a variety of deep learning models with carefully designed architectures have been proposed (Nie et al., 2022; Zhang et al., 2025b; Huang et al., 2024). These models aim to capture the underlying dynamics of MTS, such as long-term trends and periodic patterns. However, recent studies (Xu et al., 2024) suggest that unimodal deep models may have reached a performance plateau, where further improvements from increasingly complex architectures are marginal. For example, when relying solely on historical traffic data without access to weather information, unimodal models are unlikely to accurately predict the changes in traffic flow caused by upcoming heavy rain.

Traditionally, exogenous textual information, such as weather condition above, has served as an essential and even indispensable reference for manual forecasting, on which domain experts often rely to anticipate future conditions and trends (Williams et al., 2024). Considering that exogenous textual information provides complementary sources of information for TSF tasks, recent efforts (Zhang et al., 2024; Zhou et al., 2023) have been made to introduce exogenous textual information into deep

models. The emergence of LLMs has further provided a new paradigm for processing textual data, enabling TS and text, as two heterogeneous modalities, to be represented and integrated within a unified framework. This advancement opens up new possibilities for enhancing the automatic reasoning and forecasting of MTS with the aid of textual information (Zhang et al., 2024). Meanwhile, with the in-depth study and wide application of the Transformer architecture, self-attention mechanisms have been increasingly employed to model system dynamics of MTS (Chang et al., 2025).

However, existing methods and analyses for text-augmented MTSF still exhibit three major gaps or limitations: (1) **Lack of theoretical foundation.** The augmentation for TSF by textual information was initially motivated by engineering intuition and subsequently validated through empirical studies (Rodrigues et al., 2019). However, to the best of our knowledge, this augmentation still lacks a theoretical foundation based on probability theory and information theory. (2) **Inflexible TS retrieval.** Retrieving relevant TS channels and feeding them into the forecasting model can improve prediction accuracy, especially when the ratio of the output length (or forecasting horizon) to the sum of the input and output lengths is relatively high. In such cases, relevant TS retrieval becomes an indispensable component in MTSF. Some researchers (Jing et al., 2022) have proposed methods that quantify the correlations among TS channels and retrieve relevant channels based on manually predefined graph structures, which rely heavily on expert knowledge and lack flexibility. (3) **Costly manual alignment of textual and numerical data.** Ensuring temporal alignment between textual and numerical data is essential as it requires the synchronization of reported text timestamps with the corresponding numerical time steps. Existing methods rely on manual alignment of TS and text at each time step (Xu et al., 2024), which is highly labor-intensive.

To address the issue of theoretical support, this work conducts an analysis grounded in information theory and establishes the theoretical foundation for the augmenting role of textual information. To flexibly capture the correlations among TS channels to retrieve relevant TS, this work proposes a frequency-domain coherence–based retrieval method. In addition, this work introduces a semantics-driven text retrieval and alignment approach that eliminates reliance on manual efforts. Furthermore, the proposed paradigm is evaluated on a comprehensive benchmark spanning diverse domains to rigorously assess its effectiveness. The main contributions of this work are summarized as follows:

- **Theoretical Grounding of Text Augmentation.** We conduct a theoretical analysis of the augmentation effect of exogenous textual information in TSF based on information theory and machine learning principles. We demonstrate that incorporating exogenous textual information reduces the uncertainty for the forecasting accuracy and provide a solid theoretical foundation, having the situation of reliance solely on engineering intuition and empirical evidence undergo an exciting transformative moment.

- **Flexible Retrieval, Alignment, and Attention-Based MTSF.** We propose a novel coherence-based relevant TS channels retrieval method, which flexibly captures complex and time-varying dependencies in MTS channels; We also propose an embedding-based retrieval and automatic alignment method that effectively associates relevant exogenous texts with corresponding time steps in TS, alleviating the reliance on manual collection and alignment. Building upon the above approaches and employing attention mechanisms, we propose a new paradigm: Retrieval-Based Text-Augmented Multivariate Time Series Forecasting (ReTaMTSF).

- **Extensive Evaluations with Significant Improvement.** ReTaMForecaster, the baseline model for ReTaMTSF, is evaluated on a multi-domain benchmark, achieving the best or second-best performance in most experiments, with up to a 74% reduction in MSE compared to the best baseline. Further ablation studies demonstrate the augmentation effect of textual information in TSF tasks.

We include additional related works in Appendix A.

## 2 INFORMATION-THEORETIC GROUNDING OF TEXT-AUGMENTED TSF

In this section, we establish connections among the uncertainty of forecasting accuracy, MSE, and mutual information (MI) to demonstrate that incorporating relevant textual information enhances TSF performance.

## 2.1 PROBLEM FORMULATION

Let the historical TS within the look-back window be denoted as $\mathbf{x} = (x_1, x_2, ..., x_L)$; the retrieved exogenous textual information as $\mathbf{y} = (y_{L+1}, y_{L+2}, ..., y_{L+H})$; the TS generated by the TSF model as $\hat{\mathbf{x}} = (\hat{x}_{L+1}, \hat{x}_{L+2}, ..., \hat{x}_{L+H})$; the ground-truth TS over the prediction horizon as $\tilde{\mathbf{x}} = (\tilde{x}_{L+1}, \tilde{x}_{L+2}, ..., \tilde{x}_{L+H})$, where $L$ is the length of the look-back window and $H$ is the length of prediction horizon. The relations among the above variables are illustrated in Fig. 1, where $g$ and $r$ denote the generation model and the retrieval model (or method), respectively, and $p$ characterizes the relation between $\hat{\mathbf{x}}$ and $\tilde{\mathbf{x}}$. We assume that $p$ follow Gaussian distributions, which can be expressed as:

$$p(\tilde{\mathbf{x}} \mid \hat{\mathbf{x}}) = \mathcal{N}\left(\tilde{\mathbf{x}} \mid \hat{\mathbf{x}}, \sigma^2 \boldsymbol{I}\right) \qquad (1)$$

where $\mathcal{N}$ denotes the Gaussian distribution, $\sigma$ is the standard deviation, and $\boldsymbol{I} \in \mathbb{R}^{H \times H}$ is the identity matrix. The uncertainty of the accuracy of $\hat{\mathbf{x}}$, denoted as $\Delta(\hat{\mathbf{x}})$, can be quantitatively defined by conditional entropy as follows:

$$\Delta(\hat{\mathbf{x}} \mid \mathbf{m}_{ref}) = H(\tilde{\mathbf{x}} \mid \hat{\mathbf{x}} = g(\mathbf{m}_{ref})) \qquad (2)$$

where $\mathbf{m}_{ref}$ denotes the reference data or information for prediction, i.e. the TSF model input. The augmentation effect of textual information on TSF can be formally expressed as:

$$\Delta(\hat{\mathbf{x}} \mid \mathbf{x}) \geq \Delta(\hat{\mathbf{x}} \mid \mathbf{x}, \mathbf{y}) \qquad (3)$$

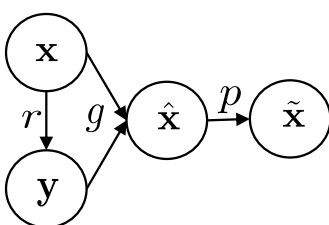

(a) With text augmentation

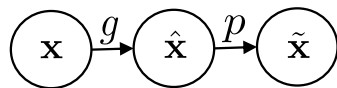

(b) Without text augmentation

Figure 1: Relations between variables for predictions w/ or w/o text augmentation. Historical TS $\mathbf{x}$ retrieves exogenous texts $\mathbf{y}$ through retrieval model (or method) $r$ in text-augmented prediction. Then generation model $g$ makes predictions $\hat{\mathbf{x}}$ based on inputs. $p$ characterizes the relation between $\hat{\mathbf{x}}$ and $\tilde{\mathbf{x}}$.

## 2.2 ANALYSIS AND PROOF

By combining the formation of information entropy and Eq. (1), the uncertainty can be expressed as:

$$\Delta(\hat{\mathbf{x}}) = H(\tilde{\mathbf{x}} \mid \hat{\mathbf{x}}) = \frac{H}{2} \log 2\pi e \sigma^2 \qquad (4)$$

where $\Delta(\hat{\mathbf{x}})$ denotes the uncertainty of the prediction accuracy when the input is unspecified and the detailed derivation is provided in the Appendix B. It can be seen that $\Delta(\hat{\mathbf{x}})$ depends only on the standard deviation $\sigma$, which can be computed from the predicted outputs $\hat{\mathbf{x}}$ and corresponding ground-truth values $\tilde{\mathbf{x}}$ as:

$$\sigma = \sqrt{\frac{1}{Z} \sum_{i=1}^{Z} (\tilde{x}_i - \hat{x}_i)^2} = \sqrt{MSE} \qquad (5)$$

where $Z$ denotes the total number of predicted-output and ground-truth element pairs. This effectively establishes an equivalence between uncertainty and MSE under the assumption of a Gaussian distribution. Meanwhile, minimizing the MSE between $\tilde{\mathbf{x}}$ and $\hat{\mathbf{x}}$ is also equivalent to maximizing the log-likelihood $\log p(\tilde{\mathbf{x}} \mid \hat{\mathbf{x}})$ (see the Appendix B for the detailed derivation), which can be expressed as:

$$\min MSE \Leftrightarrow \max \mathbb{E}_{p(\tilde{\mathbf{x}}, \hat{\mathbf{x}})}[\log p(\tilde{\mathbf{x}} \mid \hat{\mathbf{x}})] \qquad (6)$$

According to the relation between MI and entropy, we have $I(\tilde{\mathbf{x}}; \hat{\mathbf{x}}) = H(\tilde{\mathbf{x}}) - H(\tilde{\mathbf{x}} \mid \hat{\mathbf{x}})$, by treating the ground-truth value $\tilde{\mathbf{x}}$ as constants, the following formula holds:

$$\max I(\tilde{\mathbf{x}}; \hat{\mathbf{x}}) \Leftrightarrow \max -H(\tilde{\mathbf{x}} \mid \hat{\mathbf{x}}) = \max \mathbb{E}_{p(\tilde{\mathbf{x}}, \hat{\mathbf{x}})}[\log p(\tilde{\mathbf{x}} \mid \hat{\mathbf{x}})] \qquad (7)$$

The MSE serves as a bridge that establishes the connection between uncertainty and MI:

$$\min \Delta(\hat{\mathbf{x}}) \Leftrightarrow \min MSE \Leftrightarrow \max I(\tilde{\mathbf{x}}; \hat{\mathbf{x}}) \qquad (8)$$

Considering the chain rule of MI, we have $I(\tilde{\mathbf{x}}; \mathbf{x}, \mathbf{y}) = I(\tilde{\mathbf{x}}; \mathbf{x}) + I(\tilde{\mathbf{x}}; \mathbf{y} \mid \mathbf{x})$, and with $I(\tilde{\mathbf{x}}; \mathbf{y} \mid \mathbf{x}) \geq 0$, we obtain $I(\tilde{\mathbf{x}}; \mathbf{x}, \mathbf{y}) \geq I(\tilde{\mathbf{x}}; \mathbf{x})$. Assuming that the generative model $g$ is capable of integrating its inputs, we then obtain $I(\tilde{\mathbf{x}}; \hat{\mathbf{x}} = g(\mathbf{x}, \mathbf{y})) \geq I(\tilde{\mathbf{x}}; \hat{\mathbf{x}} = g(\mathbf{x}))$. Finally, by the equivalence relation, Eq. (3) is proved.

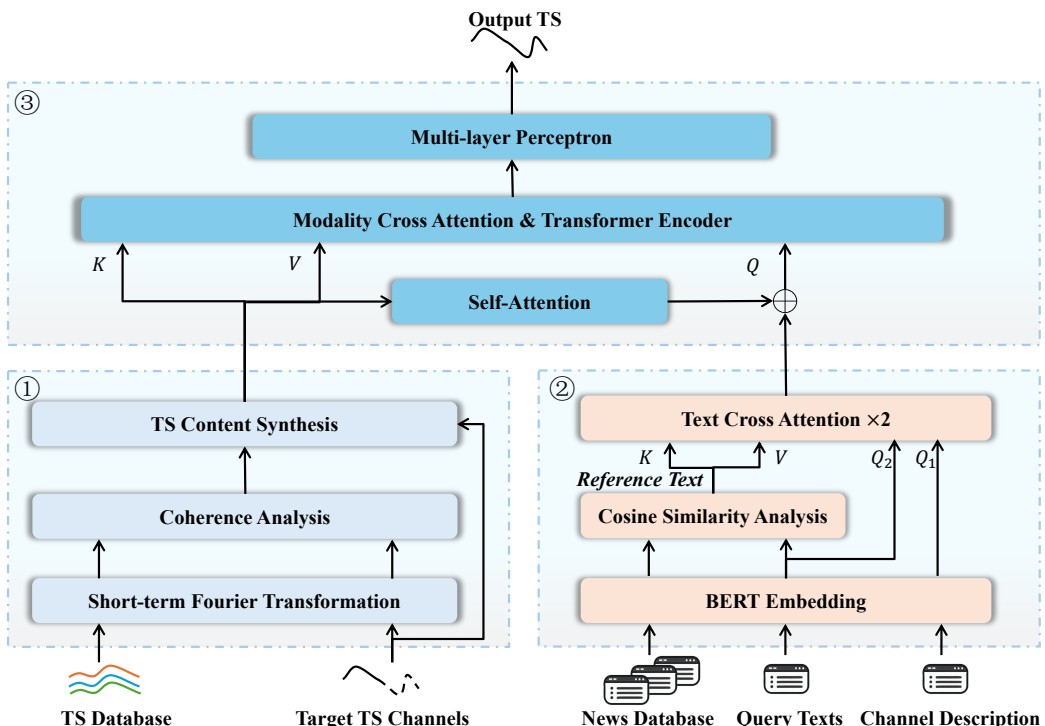

Figure 2: Model design of ReTaMForecaster. The model consists of three main modules: ① **TS retrieval and synthesis module**, which retrieves the reference TS most relevant to the target time series channels (TTC) and performs content synthesis; ② **text retrieval and alignment module**, which retrieves relevant textual information and aligns it with the TTC and temporal steps; ③ **modality alignment and output module**, which aligns the MTS with the retrieved text modality and produces the final output through Transformer-based encoding.

## 3 APPROACHES AND MODEL DESIGN

To validate the effectiveness of our proposed ReTaMTSF paradigm, we develop ReTaMForecaster, a streamlined baseline model for MTSF. As illustrated in Fig. 2, the model leverages our designed TS retrieval mechanism and exogenous text retrieval mechanism to match the most relevant TS channels and textual information as reference for the target channels and time periods. In addition, the text retrieval and alignment module enables automatic text alignment with time steps in the TS. Building on the theoretical foundation presented in section 2, the model leverages modality alignment through a cross attention mechanism to exploit textual information for enhancing the accuracy of MTSF.

### 3.1 TS RETRIEVAL AND SYNTHESIS MODULE

In MTS analysis, variables across different channels often exhibit interdependencies, i.e., inter-channel correlations. Such correlations play a crucial role in improving the accuracy of MTSF but not all channels demonstrate significant correlation with the TTC as illustrated in Fig. 3. From a mathematical perspective, TSF benchmark datasets exhibit intrinsic low-rank characteristics (Chen & Sun, 2020; Liu, 2022). Focusing the analysis only on reference TS with strong correlations to the TTC essentially corresponds to a reduced-rank regression (RRR) analysis, which not only improves computational efficiency but also mitigates the risk of potential data contamination. The inter-channel correlations of MTS in the time domain can be transformed into coherence in the frequency domain. Building on this property, we propose a reference TS retrieval approach based on coherence as illustrated in Fig. 2.

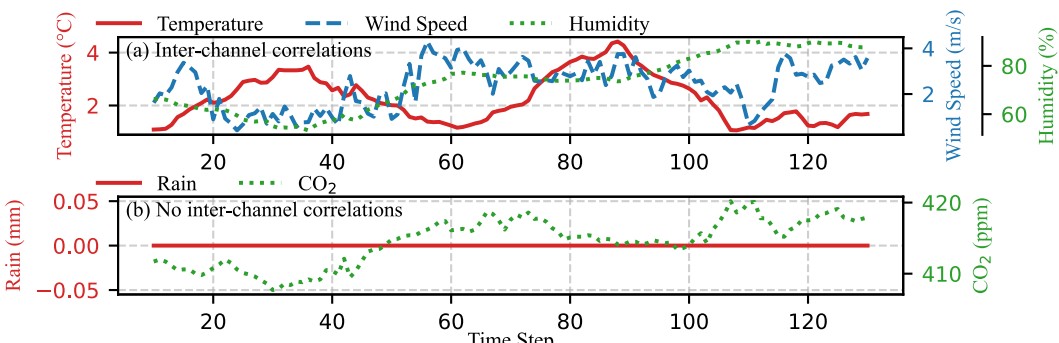

Figure 3: Example for inter-channel correlations of a weather dataset (Xu et al., 2024). Figure (a) illustrates the temporal variations of three TS variables, temperature, wind speed, and humidity, highlighting the inter-channel correlations among them. Generally, higher temperatures increase the air's moisture capacity and may lower relative humidity. Greater humidity raises air density, which can slow wind, while higher temperatures can also intensify pressure differences and generate stronger winds. Figure (b) depicts the temporal dynamics of two TS variables, rain and $CO_2$, indicating that there is almost no inter-channel correlation between them.

Since TS are discrete-time signals, we first employ the short-term fourier transform (STFT) to convert both the time series database (TSD, which includes all TS channels except TTC) $\mathbf{x}_D$ and the TTC $\mathbf{x}_T$ into the frequency domain. Subsequently, coherence analysis and retrieval are performed. The windowed STFT can be expressed as:

$$X(n, f) = \sum_{t=1}^{L} \mathbf{x}(t)\omega(t - n)e^{-ift}, \mathbf{x} = \mathbf{x}_D, \mathbf{x}_T \tag{9}$$

where $X_D, X_T$ denote all channels transformed in $\mathbf{x}_D, \mathbf{x}_T$, $n$ represents the time shift, $f$ is the frequency, and $\omega(\cdot)$ is the Hanning window function. In the subsequent coherence analysis, we compute the cross power spectral density (CPSD) between every channel in $\mathbf{x}_D$ and $\mathbf{x}_T$, as well as their respective auto power spectral densities (APSDs):

$$P_{\mathbf{x}_D^i \mathbf{x}_T^j}(f) = \mathbb{E}[X_D^i(n, f)X_T^{j*}(n, f)], \quad i = 1, 2, \ldots, C_D, \; j = 1, 2, \ldots, C_T \tag{10}$$

$$P_{\mathbf{x}^k \mathbf{x}^k}(f) = \mathbb{E}[X^k(n, f)^2], \quad \mathbf{x} = \mathbf{x}_D, \; X = X_D, \; k = i \text{ or } \mathbf{x} = \mathbf{x}_T, \; X = X_T, \; k = j \tag{11}$$

where $\mathbf{x}_D^i$ and $\mathbf{x}_T^j$ denote $i$-th channel of $\mathbf{x}_D$ and $j$-th channel of $\mathbf{x}_T$ respectively, $P_{\mathbf{x}_D^i \mathbf{x}_T^j}(f)$ represents CPSD between $\mathbf{x}_D^i$ and $\mathbf{x}_T^j$, $P_{\mathbf{x}_D^i \mathbf{x}_D^i}(f)$ and $P_{\mathbf{x}_T^j \mathbf{x}_T^j}(f)$ represent APSD for $\mathbf{x}_D^i$ and $\mathbf{x}_T^j$ respectively, $C_D$ and $C_T$ denote the number of channels in $\mathbf{x}_D$ and $\mathbf{x}_T$ respectively. $\mathbb{E}[\cdot]$ denotes the expectation operator, which in practice corresponds to averaging across windows (along the $n$-dimension). The mean coherence between $\mathbf{x}_D^i$ and $\mathbf{x}_T^j$ is then computed as:

$$\bar{C}_{\mathbf{x}_D^i \mathbf{x}_T^j} = \mathbb{E}[C_{\mathbf{x}_D^i \mathbf{x}_T^j}(f)] = \mathbb{E}\Big[\frac{\big|P_{\mathbf{x}_D^i \mathbf{x}_T^j}(f)\big|^2}{P_{\mathbf{x}_D^i \mathbf{x}_D^i}(f) \cdot P_{\mathbf{x}_T^j \mathbf{x}_T^j}(f)}\Big] \tag{12}$$

where $\bar{C}_{\mathbf{x}_D^i \mathbf{x}_T^j}$ denotes the mean coherence, and $\mathbb{E}[\cdot]$ here represents the averaging operation along the frequency ($f$) dimension. For each target channel in $\mathbf{x}_T$, we rank the channels in $\mathbf{x}_D$ according to their mean coherence with the target channel. The top-$K_T$ channels with the highest mean coherence values are then selected as the reference TS $\mathbf{x}_{\text{ref}} \in \mathbb{R}^{B \times C_T \cdot K_T \times L}$. Formally, the selected reference TS set is given by:

$$\mathcal{R}_j = \text{TopK}_{K_T}\left(\big\{\bar{C}_{\mathbf{x}_D^i \mathbf{x}_T^j} \,|\, i = 1, 2, \ldots, C_D\big\}\right) \tag{13}$$

$$\mathbf{x}_{\text{ref}}^j = \mathbf{x}_D^{\mathcal{R}_j}, j = 1, 2, \ldots, C_T \tag{14}$$

By employing the TS content synthesis module (Jing et al., 2022) (illustrated in Appendix C), we model both the inter-channel correlations and temporal dependencies between the reference TS and TTC, yielding $\boldsymbol{X}_{\text{syn}} \in \mathbb{R}^{B \times C_T \cdot (K_T + 1) \times L \times D}$.

## 3.2 TEXT RETRIEVAL AND ALIGNMENT MODULE

In natural language processing (NLP) researches, textual information is typically mapped into a semantic space of the model, i.e., transformed into text embeddings to capture semantic information and facilitate downstream tasks. In TSF, incorporating text embeddings can provide external knowledge to the model, thereby enhancing predictive accuracy (Xu et al., 2024; Jin et al., 2024b; Zhang et al., 2025a). To alleviate the burden of precisely aligning text with time steps in TS, we propose a text retrieval and alignment module as illustrated in Fig. 2.

Query texts, which include information about the prediction time steps, and channel descriptions, which describe the TTC, are designed as inputs, with detailed examples provided in Appendix D. Within the horizon window length, each time step corresponds to a query text, where the semantics of each query text encapsulate the shortest time interval associated with that step, consistent with the dataset's sampling frequency. The channel descriptions refer to a textual characterization of the channel's properties, which is generated by LLMs based on the channel name. Each channel in the MTS is associated with a channel description but in our experiments, only the channel description of the target channel is utilized. The query texts and channel descriptions are then processed together with the exogenous news database texts using BERT to obtain their embeddings, which are subsequently normalized to unit length.

Continuously, we perform cosine similarity analysis between the embeddings of the news database $\boldsymbol{Y}_{\text{nd}}$ and the embeddings of the query texts $\boldsymbol{Y}_{\text{qt}}$. Since words that are close in the semantic space tend to share similar meanings, whereas those that are distant exhibit less semantic similarity (Mikolov et al., 2013), we compute the cosine similarity between every query text and every news item in the database. By ranking the results, we retrieve the top-$K_n$ most relevant news items for each query text, which together constitute reference text embedding $\boldsymbol{Y}_{\text{ref}} \in \mathbb{R}^{B \times C_T \cdot K_n \times H \times D}$. The process is formally expressed as:

$$\mathcal{R}_j = \text{TopK}_{K_n} \left( \left\{ \cos(\boldsymbol{Y}_{\text{nd}}^i, \boldsymbol{Y}_{\text{qt}}^j) \mid i = 1, 2, \ldots, N \right\} \right)$$
$$\boldsymbol{Y}_{\text{ref}}^j = \boldsymbol{Y}_{\text{nd}}^{\mathcal{R}_j}, j = 1, 2, \ldots, C_T \cdot H \tag{15}$$

Since each time step of the TTC corresponds to a query text, and each query text retrieves $K_n$ reference texts, the task naturally involves aligning textual information with the corresponding time steps in TS instead of costly manual alignment.

Ultimately, we employ two successive cross attention modules to aggregate the textual information. In the first module, the reference text embeddings $\boldsymbol{Y}_{\text{ref}}$ are used as keys and values, while the channel description embeddings $\boldsymbol{Y}_{\text{des}}$ serve as queries, in order to calculate the relevance of each news item to every TTC and generate a composite embedding for each TTC. In the second module, the output from the first module is used as keys and values, and the query text embeddings $\boldsymbol{Y}_{\text{qt}}$ serve as queries, to calculate the relevance of the news items to each time step within the forecasting horizon, thereby culminating in a composite embedding for every time step, which can be expressed as $\boldsymbol{Y}_{\text{syn}} \in \mathbb{R}^{B \times C_T \times H \times D}$.

## 3.3 MODALITY ALIGNMENT AND OUTPUT MODULE

In this module, the fusion of TS and text modalities is achieved through cross attention mechanism, illustrated by Fig. 2. First, the last label length time steps of the output $\boldsymbol{X}_{\text{syn}}$ from the TS retrieval and synthesis module are extracted as the output guidance and concatenated with the output $\boldsymbol{Y}_{\text{syn}}$ from the text retrieval and alignment module, yielding $\boldsymbol{Y}_{\text{cat}} \in \mathbb{R}^{B \times C_T \times (L_{\text{label}} + H) \times D}$. Subsequently, $\boldsymbol{X}_{\text{syn}}$ is treated as keys and values, while $\boldsymbol{Y}_{\text{cat}}$ serves as queries to achieve modality alignment. The aligned representations are then modeled by Transformer encoders, and finally, an multi-layer perceptron (MLP) layer generates the predicted TS $\hat{\mathbf{x}} \in \mathbb{R}^{B \times C_T \times H}$.

Table 1: Overview of the selected datasets from Time-MMD benchmark.

| Domain | Target Variable | Dimension | Frequency | Number of Samples |
|---|---|---|---|---|
| Energy | Gasoline Prices | 9 | Weekly | 1479 |
| Climate | Drought Level | 6 | Monthly | 496 |
| Health (US) | Influenza Patients Proportion | 11 | Weekly | 1389 |
| Environment | Air Quality Index | 4 | Daily | 11102 |

# 4 EXPERIMENTS

Based on the proposed ReTaMForecaster baseline model, we conduct comprehensive experiments on MTSF datasets from the Time-MMD benchmark (Liu et al., 2024), thereby validating the effectiveness of the ReTaMTSF paradigm. Furthermore, we performed ablation studies to examine the contribution of textual information to augmenting MTSF performance.

## 4.1 EVALUATION ON MULTIMODAL MTSF

The Time-MMD benchmark is a multi-domain multimodal time series benchmark publicly available at `https://github.com/AdityaLab/Time-MMD`. It encompasses nine primary data domains, among which we select four multivariate datasets, namely Energy, Climate, Health (US), and Environment, to evaluate the model performance. Tab. 1 provides a overview for the four datasets we selected.

Table 2: Evaluation results on four MTSF datasets of Time-MMD. For each dataset and each horizon window length, the best result is highlighted with a gray background and the second-best result is underlined. The unimodal results of ReTaMForecaster are obtained through ablation studies detailed in section 4.2. The results of baseline models are in Liu et al. (2024).

| Dataset | | Energy | | | | Climate | | | | Health(US) | | | | Environment | | | |
|---|---|---|---|---|---|---|---|---|---|---|---|---|---|---|---|---|---|
| Horizon Window Length | | 12 | 24 | 36 | 48 | 6 | 8 | 10 | 12 | 12 | 24 | 36 | 48 | 48 | 96 | 192 | 336 |
| Model | Modal | | | | | | | | | | | | | | | | |
| FiLM | Uni | 0.21 | 0.30 | 0.37 | 0.49 | 1.42 | 1.39 | 1.40 | 1.40 | 2.53 | 2.59 | 2.46 | 2.38 | 0.32 | 0.35 | 0.35 | 0.32 |
| | Multi | 0.17 | 0.28 | 0.36 | 0.48 | 1.15 | 1.15 | 1.14 | 1.17 | 1.67 | 1.83 | 1.80 | 1.81 | 0.30 | 0.32 | 0.32 | 0.30 |
| DLinear | Uni | 0.26 | 0.32 | 0.39 | 0.50 | 1.35 | 1.41 | 1.36 | 1.36 | 2.37 | 2.61 | 2.50 | 2.48 | 0.41 | 0.57 | 0.73 | 0.59 |
| | Multi | 0.22 | 0.29 | 0.36 | 0.47 | 1.06 | 1.05 | 1.07 | 1.08 | 1.62 | 1.67 | 1.68 | 1.78 | 0.32 | 0.40 | 0.46 | 0.42 |
| Transformer | Uni | 0.18 | 0.26 | 0.36 | 0.44 | 1.04 | 1.14 | 1.12 | 1.11 | 1.22 | 1.56 | 1.43 | 1.55 | 0.32 | 0.32 | 0.48 | 0.44 |
| | Multi | 0.13 | 0.22 | 0.32 | 0.42 | 0.97 | 1.01 | 1.00 | 1.00 | 0.93 | 1.34 | 1.26 | 1.29 | 0.59 | 0.61 | 0.70 | 0.32 |
| Reformer | Uni | 0.28 | 0.38 | 0.49 | 0.57 | 1.24 | 1.06 | 1.13 | 1.16 | 1.63 | 1.99 | 1.91 | 1.90 | 0.39 | 0.45 | 0.51 | 0.48 |
| | Multi | 0.25 | 0.38 | 0.43 | 0.54 | 0.97 | 0.95 | 0.94 | 0.98 | 1.06 | 1.30 | 1.33 | 1.39 | 0.29 | 0.35 | 0.36 | 0.32 |
| Informer | Uni | 0.18 | 0.29 | 0.35 | 0.48 | 1.08 | 1.11 | 1.08 | 1.07 | 1.24 | 1.61 | 1.61 | 1.67 | 0.39 | 0.42 | 0.46 | 0.48 |
| | Multi | 0.15 | 0.24 | 0.32 | 0.44 | 1.04 | 1.03 | 1.04 | 1.02 | 0.98 | 1.23 | 1.28 | 1.40 | 0.31 | 0.33 | 0.39 | 0.34 |
| Autoformer | Uni | 0.18 | 0.31 | 0.34 | 0.47 | 1.30 | 1.24 | 1.28 | 1.25 | 1.99 | 2.25 | 2.26 | 2.39 | 0.43 | 0.36 | 0.52 | 0.37 |
| | Multi | 0.16 | 0.27 | 0.32 | 0.45 | 1.08 | 1.02 | 1.05 | 1.05 | 1.43 | 1.74 | 1.76 | 1.69 | 0.35 | 0.35 | 0.35 | 0.34 |
| FEDformer | Uni | 0.11 | 0.24 | 0.34 | 0.45 | 1.32 | 1.36 | 1.28 | 1.27 | 1.08 | 1.58 | 1.69 | 1.76 | 0.36 | 0.43 | 0.42 | 0.35 |
| | Multi | 0.09 | 0.21 | 0.32 | 0.44 | 0.98 | 1.00 | 1.03 | 1.02 | 0.92 | 1.25 | 1.36 | 1.42 | 0.30 | 0.34 | 0.34 | 0.33 |
| Nonstationary Transformer | Uni | 0.11 | 0.21 | 0.34 | 0.48 | 1.30 | 1.32 | 1.36 | 1.32 | 1.19 | 1.68 | 1.91 | 2.02 | 0.31 | 0.39 | 0.43 | 0.38 |
| | Multi | 0.10 | 0.20 | 0.28 | 0.46 | 1.00 | 1.02 | 1.02 | 1.01 | 0.94 | 1.14 | 1.17 | 1.30 | 0.29 | 0.31 | 0.32 | 0.30 |
| Crossformer | Uni | 0.14 | 0.29 | 0.36 | 0.41 | 1.12 | 1.10 | 1.12 | 1.10 | 1.45 | 1.57 | 1.62 | 1.65 | 0.34 | 0.33 | 0.73 | 0.53 |
| | Multi | 0.13 | 0.26 | 0.36 | 0.41 | 1.00 | 0.99 | 1.00 | 1.01 | 1.01 | 1.29 | 1.28 | 1.37 | 0.29 | 0.30 | 0.36 | 0.36 |
| PatchTST | Uni | 0.10 | 0.21 | 0.30 | 0.42 | 1.36 | 1.33 | 1.27 | 1.28 | 1.23 | 1.63 | 1.78 | 1.86 | 0.35 | 0.38 | 0.36 | 0.32 |
| | Multi | 0.10 | 0.21 | 0.29 | 0.41 | 0.99 | 1.01 | 1.04 | 1.06 | 0.98 | 1.27 | 1.49 | 1.60 | 0.31 | 0.32 | 0.32 | 0.30 |
| iTransformer | Uni | 0.10 | 0.21 | 0.30 | 0.42 | 1.16 | 1.23 | 1.24 | 1.22 | 1.14 | 1.62 | 1.84 | 1.89 | 0.28 | 0.29 | 0.30 | 0.28 |
| | Multi | 0.09 | 0.19 | 0.29 | 0.41 | 0.99 | 1.01 | 1.04 | 1.06 | 0.97 | 1.38 | 1.71 | 1.72 | 0.28 | 0.29 | 0.29 | 0.27 |
| Time-LLM | Uni | 0.16 | 0.27 | 0.31 | 0.45 | 1.36 | 1.26 | 1.27 | 1.27 | 1.60 | 1.94 | 1.95 | 2.17 | 0.38 | 0.37 | 0.45 | 0.33 |
| | Multi | 0.10 | 0.20 | 0.29 | 0.41 | 0.99 | 1.01 | 1.04 | 1.07 | 0.98 | 1.36 | 1.65 | 1.69 | 0.29 | 0.30 | 0.31 | 0.28 |
| ReTaM-Forecaster | Uni | 0.14 | 0.44 | 0.30 | 0.41 | 0.26 | 0.42 | 0.72 | 0.66 | 1.67 | 1.48 | 1.36 | 1.39 | 0.32 | 0.47 | 0.46 | 0.38 |
| | Multi | 0.09 | 0.18 | 0.28 | 0.39 | 0.23 | 0.40 | 0.70 | 0.61 | 0.89 | 1.18 | 1.19 | 1.34 | 0.29 | 0.30 | 0.37 | 0.35 |

We follow the general experimental setup of Time-MMD to forecast a single TTC from MTS inputs, treating the remaining channels as covariates. The horizon window lengths span from short- to long-term forecasting tasks, with four horizon window lengths for each dataset determined by its sampling frequency. We evaluate the performance of ReTaMForecaster on four MTSF datasets from Time-MMD as we mentioned above, and compare it against competitive baselines. To process textual inputs, we employ the paraphrase-MiniLM-L6-v2 model to obtain their embeddings. Model performance is assessed using the widely adopted MSE metric, where lower values indicate better predictive accuracy. As demonstrated in Tab. 2, ReTaMForecaster achieves state-of-the-art

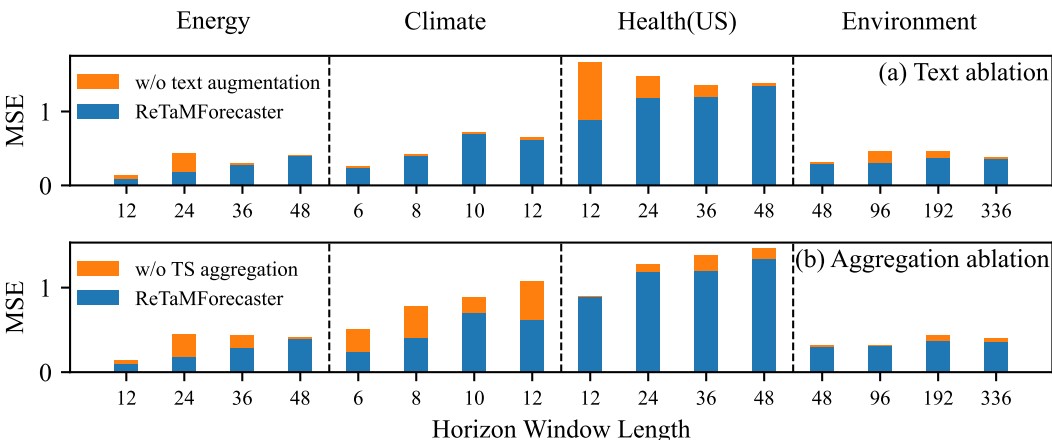

Figure 4: Results of ablation studies. Figure (a) illustrates the results of the text ablation experiment, while Figure (b) shows results of the aggregation ablation experiment. In both figures, the blue bars represent the performance of ReTaMForecaster, and the orange bars indicate the differences between the ablation experiment results and the ReTaMForecaster results, reflecting the contribution of the corresponding module.

performance across all evaluated horizon window lengths on the Energy and Climate datasets. For the Health (US) and Environment datasets, it attains the best or second-best performance in at least half of the evaluated horizon window lengths. This validates the effectiveness of ReTaMTSF, including TS and text retrieval and modality alignment through cross attention. Despite the strong performance of the multimodal version, the unimodal results of ReTaMForecaster are generally inferior to those of the baseline unimodal versions. However, the incorporation of textual information substantially enhances forecasting accuracy, with the maximum improvement achieving up to a 74% reduction in MSE compared to the best-performing baseline model. This observation is consistent with the findings of Zhang et al. (2025a), which suggest that incorporating extra modalities is particularly beneficial for weaker unimodal forecasting models as text information provides the most value when the TS model lacks sufficient capacity to capture temporal patterns on its own.

## 4.2 ABLATION STUDIES

To assess the contribution of textual information to MTSF and the role of the aggregation module in TS content synthesis, we conducted additional ablation studies. In the text ablation experiment, the composite text embedding $Y_{\text{syn}}$ is replaced with an all-zero tensor, thereby removing the augmentation effect of textual information. In the aggregation ablation experiment, the aggregation module is removed, leaving no dedicated mechanism to capture inter-channel correlations and temporal dependencies in the MTS. Fig. 4 presents the results, with detailed outcomes provided in Tab. 3.

Table 3: Ablation study results of ReTaMForecaster on four datasets with different horizon window lengths.

| Dataset | Energy | | | | Climate | | | | Health(US) | | | | Environment | | | |
|---|---|---|---|---|---|---|---|---|---|---|---|---|---|---|---|---|
| Horizon Length | 12 | 24 | 36 | 48 | 6 | 8 | 10 | 12 | 12 | 24 | 36 | 48 | 48 | 96 | 192 | 336 |
| ReTaMForecaster | 0.09 | 0.18 | 0.28 | 0.39 | 0.23 | 0.40 | 0.70 | 0.61 | 0.89 | 1.18 | 1.19 | 1.34 | 0.29 | 0.30 | 0.37 | 0.35 |
| w/o text augmentation | 0.14 | 0.44 | 0.30 | 0.41 | 0.26 | 0.42 | 0.72 | 0.66 | 1.67 | 1.48 | 1.36 | 1.39 | 0.32 | 0.47 | 0.46 | 0.38 |
| w/o TS aggregation | 0.14 | 0.45 | 0.43 | 0.41 | 0.51 | 0.78 | 0.89 | 1.07 | 0.90 | 1.27 | 1.38 | 1.46 | 0.32 | 0.32 | 0.44 | 0.40 |

Quantitatively, textual information contributes on average 18% to the performance of ReTaMForecaster, while the aggregation module accounts for an average of 23% of the observed performance improvement. The results substantiate the efficacy of incorporating exogenous textual information in addressing the information insufficiency inherent in TSF models (Xu et al., 2024), thereby val-

idating the theoretical analysis presented in section 2 and demonstrating the effectiveness of the proposed aggregation module.

### 4.3 Effect of TS and Text Retrieval

The role of TS and text retrieval in the model is investigated by examining the impact of the number of retrieved TS ($K_T$) as well as exogenous texts ($K_n$) on model performance. For each dataset, we select the shortest horizon window length for experimentation, and the results are shown in Fig. 5. It can be observed that the MSE initially decreases as $K_T$ and $K_n$ increase, reaching a minimum point. Beyond this point, however, the MSE ceases to decrease or even starts to rise, in some cases exceeding the performance of the setting without TS and text retrieval. This indicates that while MTS covariates and exogenous texts can provide additional information to improve forecasting accuracy, redundant information may introduce noise, thereby degrading model performance.

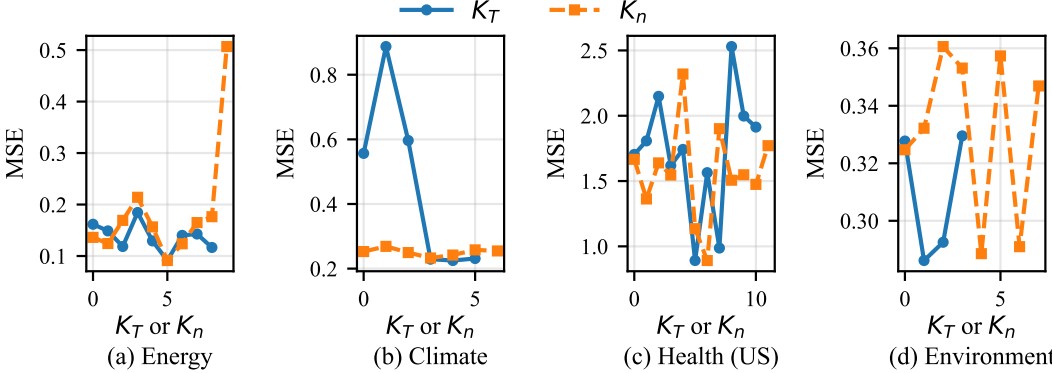

Figure 5: The impact of the number of retrieved TS ($K_T$) as well as exogenous texts ($K_n$) on model performance is investigated on (a) Energy, (b) Climate, (c) Health (US) and (d) Environment datasets.

## 5 Conclusion and Limitations

**Conclusion.** In this work, we provide a theoretical foundation for the augmentation of TSF through textual information, and subsequently propose ReTaMTSF, a retrieval-based text-augmented multivariate time series forecasting paradigm. We further design ReTaMForecaster, a baseline model for ReTaMTSF, which leverages flexible retrieval of relevant TS and exogenous text to facilitate multimodal MTSF. Extensive experiments on four multimodal MTSF datasets from Time-MMD across diverse domains demonstrate the effectiveness of ReTaMTSF, underscoring the importance of incorporating exogenous textual information and flexibly capturing inter-channel correlations. This work alleviates the manual burden inherent in prior multimodal TSF approaches and offers new perspectives and methodologies for advancing and optimizing MTSF. For future work, incorporating spatial information into the model to support spatiotemporal forecasting could provide more comprehensive predictive insights.

**Limitations.** Despite the contributions, this work still has several limitations. The theoretical analysis provided in this work regarding the augmentation effect of textual information on TSF is based on the Gaussian distribution assumption. However, the relation between ground-truth and predicted values may not necessarily follow this assumption, and further validation is required under alternative conditions. The proposed TS retrieval mechanism relies on frequency-domain coherence, which only captures linear correlations and thus fails to reflect nonlinear dependencies. In addition, when retrieving and aligning relevant textual information, the model must load a large number of text embeddings, resulting in a space complexity approximately linear in the sum of the look-back window length and the horizon window length, i.e., $\mathcal{O}(L + H)$. Consequently, long-term forecasting may incur substantial computational costs.

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

## A  RELATED WORKS

### A.1  MULTIMODAL TRANSFER LEARNING WITH LARGE LANGUAGE MODELS AND THEIR APPLICATIONS IN TIME SERIES ANALYSIS

LLMs have demonstrated remarkable performance in multimodal transfer learning, including tasks involving images (Lin et al., 2024), audio (Ghosal et al., 2023), tabular data (Hegselmann et al., 2023), and time series data (Zhou et al., 2023). A key motivation for employing LLMs in multimodal tasks is their ability to achieve strong performance even under limited data scenarios (Zhou et al., 2023). To preserve their data-independent representation learning capability, most parameters of these models are typically kept frozen, and empirical evidence suggests that LLMs with largely frozen parameters often outperform those trained from scratch(Lin et al., 2024; Zhou et al., 2023).

Current approaches for transferring and extracting the knowledge stored in LLMs parameters for TS analysis can be broadly categorized into five types (Zhang et al., 2024): (1) prompting (input

stage); (2) time series quantization (tokenization stage); (3) aligning (embedding stage); (4) vision as bridge (LLM stage); and (5) tool integration (output stage). All these methods focus on transferring the knowledge embedded in LLMs parameters to other modalities. Among these, aligning requires synchronizing textual and sequential data at the timestep level before modality alignment. However, existing methods often rely on manual alignment (Xu et al., 2024), which is both time-consuming and labor-intensive. To address this limitation, our work introduces a text retrieval and alignment mechanism that enables this process to be performed automatically, thereby improving efficiency.

## A.2 Time Series Representation Learning and Forecasting

In the TS domain, self-supervised learning has emerged as an important approach for representation learning. Although Transformers are widely recognized as a leading solution for end-to-end TS analysis (Nie et al., 2023), backbone networks based on CNNs (Yue et al., 2022) or RNNs (Tonekaboni et al., 2021) have traditionally been the preferred architectures for self-supervised learning in TS. Conventional TS forecasting methods take a statistical perspective, treating forecasting as a standard regression problem with time-varying parameters (Zhang, 2003). Recent advances in deep learning, however, have led to significant breakthroughs, giving rise to models such as LSTNet (Lai et al., 2018) and N-BEATS (Oreshkin et al., 2020).

Due to the inherent ability of the Transformer's self-attention mechanism to capture long-range dependencies and complex patterns, it is particularly well-suited for TS data with intricate sequential relationships. Consequently, many state-of-the-art deep learning methods are built upon Transformer architectures (Zhou et al., 2021; Wu et al., 2021). However, these methods overlook the low-rank property of TS datasets and rely entirely on Transformers to model the inter-channel correlations of MTS. These approaches can lead to low computational efficiency and difficulty in focusing attention on the truly relevant channels. In contrast, our work first employs numerical methods to flexibly capture the inter-channel correlations of MTS and performs dimensionality reduction accordingly, before leveraging a Transformer to model the data. This two-step approach improves both the predictive accuracy and efficiency of the model.

## A.3 Retrieval-Augmented Generation Models

Retrieval-Augmented Generation (RAG) is an emerging hybrid architecture designed to address the limitations of pure generative models. RAG integrates two key components: a retrieval mechanism, which searches for relevant documents or information from external knowledge sources, and a generation module, which processes the retrieved information to produce more accurate outputs, often in a human-like textual form. This combination enables RAG models not only to generate coherent and fluent text but also to incorporate up-to-date real-world knowledge into their outputs. For instance, Re[3]Sum (Cao et al., 2018) generates document summaries based on retrieved templates, while Song et al. (2018) suggests generating dialogue responses grounded in retrieved references. In the field of MTSF, Jing et al. (2022) introduces a retrieval-based forecasting model; however, this model relies solely on manually predefined relationships for retrieval within TS data. In contrast, our work performs flexible retrieval jointly over both TS and textual information, thereby enabling a more context-aware forecasting framework.

# B Theoretical Grounding Supplement

## B.1 Derivation of the Entropy-Based Uncertainty Expression

For a $c$-dimensional multivariate Gaussian distribution $\mathcal{N}(\boldsymbol{\mu}, \boldsymbol{\Sigma})$, the probability density function is given by:

$$p(\mathbf{x}) = \frac{1}{(2\pi)^{\frac{c}{2}} |\boldsymbol{\Sigma}|^{\frac{1}{2}}} \exp\left(-\frac{1}{2}(\mathbf{x} - \boldsymbol{\mu})^{\top} \boldsymbol{\Sigma}^{-1}(\mathbf{x} - \boldsymbol{\mu})\right) \tag{16}$$

Taking the logarithm yields:

$$\log p(\mathbf{x}) = -\frac{c}{2}\log(2\pi) - \frac{1}{2}\log|\boldsymbol{\Sigma}| - \frac{1}{2}(\mathbf{x} - \boldsymbol{\mu})^{\top}\boldsymbol{\Sigma}^{-1}(\mathbf{x} - \boldsymbol{\mu}) \tag{17}$$

Substituting into the differential entropy formula $H(\mathbf{x}) = -\int p(\mathbf{x}) \log p(\mathbf{x}) d\mathbf{x}$ gives:

$$H(\mathbf{x}) = \int p(\mathbf{x}) \left[ \frac{c}{2} \log(2\pi) + \frac{1}{2} \log |\mathbf{\Sigma}| + \frac{1}{2} (\mathbf{x} - \boldsymbol{\mu})^\top \mathbf{\Sigma}^{-1} (\mathbf{x} - \boldsymbol{\mu}) \right] d\mathbf{x} \qquad (18)$$

$$= \frac{c}{2} \log(2\pi) + \frac{1}{2} \log |\mathbf{\Sigma}| + \frac{1}{2} \mathbb{E} \left[ (\mathbf{x} - \boldsymbol{\mu})^\top \mathbf{\Sigma}^{-1} (\mathbf{x} - \boldsymbol{\mu}) \right] \qquad (19)$$

The last term can be rewritten as:

$$\frac{1}{2} \mathbb{E} \left[ (\mathbf{x} - \boldsymbol{\mu})^\top \mathbf{\Sigma}^{-1} (\mathbf{x} - \boldsymbol{\mu}) \right] = \frac{1}{2} \operatorname{tr} \left( \mathbf{\Sigma}^{-1} \mathbb{E} \left[ (\mathbf{x} - \boldsymbol{\mu})(\mathbf{x} - \boldsymbol{\mu})^\top \right] \right) = \frac{1}{2} \operatorname{tr} \left( \mathbf{\Sigma}^{-1} \mathbf{\Sigma} \right) = \frac{1}{2} c \quad (20)$$

Substituting back into Eq. 19 gives:

$$H(\mathbf{x}) = \frac{c}{2}(1 + \log 2\pi) + \frac{1}{2} \log |\mathbf{\Sigma}| \qquad (21)$$

Considering Eq. 1 gives:

$$H(\tilde{\mathbf{x}} \mid \hat{\mathbf{x}}) = \frac{c}{2}(1 + \log 2\pi) + \frac{1}{2} \log \left( \sigma^2 \right)^c = \frac{c}{2} \log 2\pi e \sigma^2 \qquad (22)$$

### B.2 Proof of the MSE–Log-Likelihood Equivalence

Assume a forecasting model as:

$$\tilde{\mathbf{x}} = f(\mathbf{x}; \boldsymbol{\theta}) + \boldsymbol{\epsilon} = \hat{\mathbf{x}} + \boldsymbol{\epsilon}, \boldsymbol{\epsilon} \sim \mathcal{N}\left(0, \sigma^2 \boldsymbol{I}\right) \qquad (23)$$

where $\mathbf{x}$ denotes the input sequence to the model, $\hat{\mathbf{x}}$ represents the predicted sequence, $\tilde{\mathbf{x}}$ denotes the ground-truth sequence over the forecasting horizon, $\boldsymbol{\theta}$ corresponds to the model parameters, and the distribution function of $\boldsymbol{\epsilon}$ is given by:

$$p(\boldsymbol{\epsilon}) = \frac{1}{(2\pi\sigma^2)^{\frac{c}{2}}} \exp \left( -\frac{\|\boldsymbol{\epsilon}\|^2}{2\sigma^2} \right) \qquad (24)$$

where $c$ denotes the dimension of the sequence. Then the conditional distribution of $\tilde{\mathbf{x}}$ can be expressed as:

$$p(\tilde{\mathbf{x}} \mid \hat{\mathbf{x}}) = p(\tilde{\mathbf{x}} \mid f(\mathbf{x}; \boldsymbol{\theta})) = \mathcal{N}\left(\tilde{\mathbf{x}} \mid f(\mathbf{x}; \boldsymbol{\theta}), \sigma^2 \boldsymbol{I}\right) = \frac{1}{(2\pi\sigma^2)^{\frac{c}{2}}} \exp \left( -\frac{\|\tilde{\mathbf{x}} - \hat{\mathbf{x}}\|^2}{2\sigma^2} \right) \qquad (25)$$

Given a dataset $\mathcal{D} = \{(\mathbf{x}_i, \tilde{\mathbf{x}}_i)\}_{i=1}^N$, $\mathbf{x}_i$ and $\tilde{\mathbf{x}}_i$ denote the values at the $i$-th time step in the input sequence and the ground-truth sequence, respectively, the likelihood function can be written as:

$$p(\mathcal{D} \mid \boldsymbol{\theta}) = \prod_{i=1}^N p\left(\tilde{\mathbf{x}}_i \mid \mathbf{x}_i; \boldsymbol{\theta}\right) = \prod_{i=1}^N p\left(\tilde{\mathbf{x}}_i \mid \hat{\mathbf{x}}_i\right) \qquad (26)$$

Taking the logarithm of the likelihood function, we obtain the log-likelihood function:

$$\log p(\mathcal{D} \mid \boldsymbol{\theta}) = \sum_{i=1}^N \log p\left(\tilde{\mathbf{x}}_i \mid \hat{\mathbf{x}}_i\right) \qquad (27)$$

Considering Eq. 1 gives:

$$\sum_{i=1}^N \log p\left(\tilde{\mathbf{x}}_i \mid \hat{\mathbf{x}}_i\right) = \sum_{i=1}^N \left[ -\frac{c}{2} \log \left(2\pi\sigma^2\right) - \frac{\|\tilde{\mathbf{x}}_i - \hat{\mathbf{x}}_i\|^2}{2\sigma^2} \right] \propto -\frac{1}{2\sigma^2} \sum_{i=1}^N \|\tilde{\mathbf{x}}_i - \hat{\mathbf{x}}_i\|^2 \qquad (28)$$

MSE is defined as $\frac{1}{N} \sum_{i=1}^N \|\tilde{\mathbf{x}}_i - \hat{\mathbf{x}}_i\|^2$ and we can obtain:

$$\min_{\boldsymbol{\theta}} MSE \Leftrightarrow \min_{\boldsymbol{\theta}} \Sigma \|\tilde{\mathbf{x}}_i - \hat{\mathbf{x}}_i\|^2 \Leftrightarrow \max_{\boldsymbol{\theta}} \mathbb{E}_{p(\tilde{\mathbf{x}}, \hat{\mathbf{x}})}[\log p(\tilde{\mathbf{x}} \mid \hat{\mathbf{x}})] \qquad (29)$$

## C  TS CONTENT SYNTHESIS

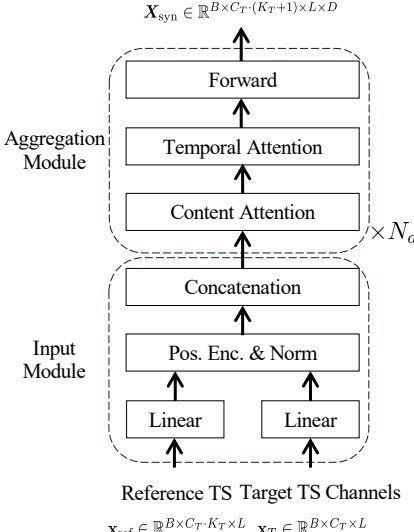

Figure 6: The architecture of the TS content synthesis module.

As illustrated in Fig. 6, the architecture of the TS content synthesis module consists of an input module and aggregation modules. In the input module, the reference TS and TTC are individually processed by a linear layer, followed by positional encoding and normalization. The resulting representations are then concatenated and passed into $N_a$ successive aggregation modules. Within each aggregation module, content attention and temporal attention are employed to capture inter-channel correlations and temporal dependencies. Finally, the representations are refined through a feed-forward network, producing the synthesized output $\boldsymbol{X}_{\mathrm{syn}}$.

## D  QUERY TEXT AND CHANNEL DESCRIPTION EXAMPLES

### D.1  QUERY TEXT

> **Example**: In the weekly reported Energy dataset, the query text corresponding to April 5, 1993, is:
>
> "From 1993-04-05 to 1993-04-11."

### D.2  CHANNEL DESCRIPTION

> **Example**: In the Energy dataset, the channel description for target channel "Gasoline Prices" is:
>
> "Gasoline prices refer to the retail cost consumers pay per gallon of gasoline, reflecting factors such as crude oil prices, taxes, and refining costs."

## E  THE USE OF LARGE LANGUAGE MODELS (LLMs)

Large Language Models (LLMs) were employed exclusively to assist with the writing process. Specifically, they were used to polish the language, improve grammar, and enhance clarity of pre-

sentation. The LLMs did not contribute to research ideation, methodology design, data analysis, or interpretation of the results.

