# OpenReview forum: "ReTaMTSF: Retrieval-Based Multimodal Framework for Multivariate Time Series Forecasting"
_ICLR.cc/2026/Conference — ICLR 2026 Conference Withdrawn Submission_

### Official Review · Reviewer_AEQ2 · 2025-10-30

**Soundness:** 4
**Presentation:** 3
**Contribution:** 4
**Rating:** 8
**Confidence:** 5

**Summary:**

This paper proposes ReTaMTSF, a retrieval-based multimodal framework for multivariate time series forecasting, integrating exogenous textual information with numerical time series data. It establishes an information-theoretic foundation demonstrating that incorporating text reduces forecasting uncertainty, connecting mutual information with mean squared error.

The authors further introduce ReTaMForecaster, a baseline model that (1) retrieves correlated time series channels using frequency-domain coherence, (2) performs semantic text retrieval and automatic temporal alignment via BERT embeddings and cross attention, and (3) fuses multimodal features for prediction.

Extensive experiments on the Time-MMD benchmark (Energy, Climate, Health, and Environment domains) show strong results. ReTaMForecaster achieves state-of-the-art or second-best performance in most cases, reducing MSE by up to 74% compared to strong baselines.

**Strengths:**

1.	Provides a rigorous information-theoretic explanation of why exogenous text improves forecasting performance, which is a rare and valuable contribution in multimodal TSF research.

2.	Introduces frequency-domain coherence for dynamic retrieval of relevant time series channels and semantic text alignment without manual labeling.

3.	Extensive experiments across multiple domains demonstrate strong and consistent improvements.

4.	Writing quality and visual presentation are good, making the idea easy to follow.

**Weaknesses:**

1. The use of MiniLM embeddings is reasonable but fixed.

2. An ablation comparing embedding models could provide more general insights.

**Questions:**

1.	How sensitive is the model to the choice of LM?

2.	What is the approximate runtime overhead of text retrieval compared to unimodal baselines on large datasets?

---

### Official Review · Reviewer_YJyz · 2025-10-31

**Soundness:** 3
**Presentation:** 3
**Contribution:** 2
**Rating:** 2
**Confidence:** 4

**Summary:**

This paper, ReTaMTSF, proposes a retrieval-based multimodal framework for multivariate time series forecasting (MTSF). The key idea is to augment time series data with external textual information retrieved from a news database. The authors provide an information-theoretic justification—showing the equivalence between mutual information (MI), mean squared error (MSE), and entropy—to argue that text augmentation theoretically reduces predictive uncertainty. The model incorporates a dual cross-attention mechanism (channel-level and time-level) to align relevant textual information with the target channel (TTC) and time horizon, and then fuses time series and text features through modality cross-attention and Transformer encoders. The paper reports improved forecasting accuracy across multiple datasets compared to strong baselines.

**Strengths:**

The paper provides a clear information-theoretic grounding for text augmentation by leveraging the MI--MSE--entropy equivalence. Specifically, it argues that when the retrieved text increases the conditional mutual information \(I(\tilde{x}; y \mid x)\), predictive uncertainty (entropy) is reduced, thereby yielding a lower MSE. This moves beyond heuristic intuition and establishes a principled justification for incorporating textual signals.

**Weaknesses:**

- The paper should additionally evaluate time-domain channel correlations (e.g., lagged/partial correlations, Granger-type analysis) to provide a quantitative comparison against the proposed frequency-domain coherence–based channel selection approach.
- The model integrates text and time-series information using a relatively simple attention mechanism; however, due to potential cross-modal misalignment between text and time-series signals, it remains uncertain whether this fusion strategy ensures effective and reliable information alignment across modalities..

**Questions:**

Because retrieval is driven by cosine similarity, the "reference text" is likely to be paraphrastic and thus reiterate information already encoded in the time series \(x\), rather than introduce truly exogenous or novel signals. As a result, the paper does not substantiate that the retrieved text increases conditional information—i.e., \(I(\tilde{x}; y \mid x) > 0\)—beyond what \(x\) already provides. Clear evidence is needed to guarantee the informativeness and quality of the retrieved text (e.g., controlled ablations).

---

### Official Review · Reviewer_jkz7 · 2025-11-01

**Soundness:** 2
**Presentation:** 2
**Contribution:** 2
**Rating:** 4
**Confidence:** 3

**Summary:**

The paper proposes ReTaMTSF, a retrieval-based, text-augmented MTSF paradigm, with a baseline ReTaMForecaster that retrieves correlated TS channels (frequency-domain coherence), retrieves/alines exogenous texts (BERT embeddings), and fuses modalities via cross-attention. Theoretically, it claims that adding relevant text increases MI and reduces uncertainty/MSE; empirically, it evaluates on four Time-MMD datasets with ablations

**Strengths:**

- Clear problem framing of multimodal TSF and reduction of manual text–time alignment via a semantics-driven retrieval/alignment module.

**Weaknesses:**

- The information-theoretic argument reads as quite intuitive (“more relevant text -> higher MI -> lower error”). I’m curious what non-obvious takeaway the theory adds, and how far it extends beyond the stated assumptions.
- Architecturally, the main differentiator appears to be the retrieval pipeline, while fusion relies on fairly standard cross-attention. Is there a clearer statement of what’s novel in the end-to-end design?
- Reported improvements over strong multimodal baselines seem to vary by domain/horizon. Would per-dataset effect sizes (beyond horizon snapshots) help demonstrate consistency?
- On the MI<->MSE link, the analysis seems to lean on Gaussian-like noise. In settings with heavier tails or heteroscedasticity (common in MTSF), how robust are the conclusions? Any stress tests you could share?
- For coherence-based retrieval, since coherence is a linear/global measure, could phase lags, transient bursts, or nonlinear relations reduce retrieval quality? A comparison with nonlinear or representation-based retrieval might clarify this.

**Questions:**

- Can you give absolute overhead numbers (forward time, peak RAM, GPU hours) vs. strong unimodal and multimodal baselines across H and K? The O(L+H) text-embedding storage suggests real costs at scale
- Given linear-coherence retrieval, what happens when relevant signals are phase-shifted or transient (burst-like) so coherence is diluted? A targeted stress test would clarify retrieval fidelity.

---

### Official Review · Reviewer_WT5k · 2025-11-01

**Soundness:** 2
**Presentation:** 2
**Contribution:** 2
**Rating:** 2
**Confidence:** 4

**Summary:**

This paper proposes ReTaMTSF, a retrieval-augmented framework for multivariate time series forecasting (MTSF) that aims to incorporate exogenous text. The authors claim contributions of an information-theoretic analysis to provide a theoretical grounding for text-augmented forecasting; a new paradigm, ReTaMTSF, with a baseline model (ReTaMForecaster) that retrieves relevant time series channels using coherence analysis and relevant text using semantic similarity, thereby reducing the need for manual text alignment and State-of-the-art experimental results on four datasets from the Time-MMD benchmark. While the problem of automatically aligning and integrating textual information into time series forecasting is important, the paper suffers from several fundamental flaws in its theoretical claims, experimental design, and comparison to prior work.

**Strengths:**

- The paper addresses a significant and practical problem. Manually collecting and aligning multimodal data (especially text) with time series is a major bottleneck, and an effective retrieval-based solution would be a valuable contribution.

- On the selected benchmarks and against the chosen baselines, the proposed ReTaMForecaster model shows strong performance, achieving SOTA or near-SOTA results in many cases (Table 2).

**Weaknesses:**

1. Vague and Potentially Misleading Experimental Setup

This is the most critical flaw. The paper's core premise (Section 3.2) is to "alleviate the burden of precisely aligning text" by retrieving relevant information from a large, unaligned "News Database" using semantic queries.

However, the experiments are conducted on the Time-MMD benchmark (Liu et al., 2024), which is a benchmark specifically designed to provide already-aligned text and other modalities. The paper provides zero details on the "News Database" and "TS Database" used in the experiments.

This raises critical, unanswered questions:

- Did the authors use a separate, large-scale, unaligned news corpus? If so, what was it? How was it curated? What was its scale? This information is completely missing.

- Or, did the authors simply use the pre-aligned text provided by the Time-MMD benchmark? If this is the case, the entire text retrieval contribution (Section 3.2) is invalidated, as the model would not be performing "retrieval and alignment" but rather just using data that was already manually aligned.

This ambiguity undermines the paper's central claim. The method is presented as a solution for unaligned data, but the experiment seems to use aligned data, making the retrieval component's purpose unclear.

2. Missing Key Baseline Comparisons

The paper positions itself as a retrieval-based forecasting model and explicitly borrows the "TS content synthesis module" from Jing et al. (2022) (cited in Section 3.1 and Appendix A.3). The authors even state their work is an improvement over models like Jing et al. (2022), which they claim "relies solely on manually predefined relationships" (lines 689-691). Further, the model architecture looks resemble to the TGTSF model proposed in the cited paper Xu et al. 2024.

Despite building on this foundation and claiming superiority, Jing et al. (2022) and Xu et al. 2024 is conspicuously absent from the experimental comparison in Table 2. This is a major omission. Without these baseline, it is impossible to assess whether the proposed coherence-based TS retrieval and text-based retrieval offer any real advantage over the prior art they are directly related to.

3. Weak and Poorly-Connected Theoretical Contribution

The information-theoretic proof in Section 2 seems irrelevant with following part although it is clear and correct.

Lack of Novelty: The proof essentially concludes that adding more relevant information (text) can increase the mutual information between the prediction and the ground truth, which in turn reduces MSE. This is a well-known and intuitive concept, not a significant theoretical contribution.

Irrelevance to Model Design: The proof relies on strong, simplifying assumptions (e.g., Gaussian distributions) and provides no concrete guidance for the specific architectural design of the ReTaMForecaster. It serves as a high-level justification for "using text" but does not inform how to retrieve or fuse it, which is the supposed contribution of the paper.

**Questions:**

See the weaknesses.

---

### Note · Authors · 2026-01-13

I have read and agree with the venue's withdrawal policy on behalf of myself and my co-authors.